# Developing artificial intelligence tools for institutional review board pre-review: A pilot study on ChatGPT's accuracy and reproducibility

Yasuko Fukataki[1,2]*, Wakako Hayashi[1], Naoki Nishimoto[3], Yoichi M. Ito[1,3]

**1** Health Data Science, Department of Social Science, Graduate School of Medicine, Hokkaido University, Sapporo, Japan, **2** Biostatistics and Data Management, Sapporo Medical University, Sapporo, Japan, **3** Data Science Center, Promotion Unit, Institute of Health Science Innovation for Medical Care, Hokkaido University Hospital, Sapporo, Japan

* yafukataki@sapmed.ac.jp

## Abstract

This pilot study is the first phase of a broader project aimed at developing an explainable artificial intelligence (AI) tool to support the ethical evaluation of Japanese-language clinical research documents. The tool is explicitly not intended to assist document drafting. We assessed the baseline performance of generative AI—Generative Pre-trained Transformer (GPT)-4 and GPT-4o—in analyzing clinical research protocols and informed consent forms (ICFs). The goal was to determine whether these models could accurately and consistently extract ethically relevant information, including the research objectives and background, research design, and participant-related risks and benefits. First, we compared the performance of GPT-4 and GPT-4o using custom agents developed via OpenAI's Custom GPT functionality (hereafter "GPTs"). Then, using GPT-4o alone, we compared outputs generated by GPTs optimized with customized Japanese prompts to those generated by standard prompts. GPT-4o achieved 80% agreement in extracting research objectives and background and 100% in extracting research design, while both models demonstrated high reproducibility across ten trials. GPTs with customized prompts produced more accurate and consistent outputs than standard prompts. This study suggests the potential utility of generative AI in pre-institutional review board (IRB) review tasks; it also provides foundational data for future validation and standardization efforts involving retrieval-augmented generation and fine-tuning. Importantly, this tool is intended not to automate ethical review but rather to support IRB decision-making. Limitations include the absence of gold standard reference data, reliance on a single evaluator, lack of convergence and inter-rater reliability analysis, and the inability of AI to substitute for in-person elements such as site visits.

**Data availability statement:** All data underlying the findings in this study are fully available in the Supporting Information files (S1 to S11 Files). These include source texts, GPT outputs, evaluation tables, and the mock IRB review form. No personal or identifying information is included. S1 to S10 Files are available from Figshare at: https://doi.org/10.6084/m9.figshare.28737845 S11 File (Mock IRB Evaluation Form) is available from Figshare at: https://doi.org/10.6084/m9.figshare.28838291

**Funding:** The author(s) received no specific funding for this work.

**Competing interests:** The authors have declared that no competing interests exist.

## Author summary

In this pilot study, we examined whether ChatGPT could accurately and consistently extract key elements from clinical research protocols and informed consent forms written in Japanese. These elements include the research objectives and background, research design, and participant-related risks and benefits, all essential for ethical review. This study is part of a larger project aiming to develop an explainable AI tool to assist institutional review board (IRB) members in evaluating clinical research documents, but not to automate ethical review. As a prerequisite for implementing advanced technologies such as document retrieval and task-specific adaptation, this study evaluated the basic capabilities of current large language models. We compared standard and customized prompts for GPT-4 and GPT-4o and found that GPT-4o, when guided by customized prompts, produced more accurate and consistent outputs. However, study limitations remain, including the absence of gold standard reference data, reliance on a single evaluator, and lack of convergence and inter-rater reliability analysis. This study offers important early insights into how generative AI could assist the ethical evaluation of clinical research documents written in Japanese.

## Introduction

An ethical review of clinical research is essential to ensure both participant safety and research reliability. Committees responsible for ethical review evaluate the appropriateness of study protocols and establish standards to safeguard human rights and safety. However, the review processes of institutional review boards (IRBs) worldwide, as well as certified review boards (CRBs) in Japan, face several persistent challenges. For instance, review comments from Japan's CRBs show substantial variation in terms of level of detail, structural organization, and evaluative consistency. Some CRBs use free-form formats, in which statements may be extremely brief (e.g., "no issues") or reflect differing evaluative perspectives and address different elements, even among reviewers in the same field. This variability stems from the absence of a uniform structure. In contrast, other CRBs produce reviews that are clearly organized by key elements such as research objectives, study design, and risk–benefit assessment. These disparities have prompted calls to standardize and harmonize the quality of ethical reviews [1–7]. Across different ethics review committees internationally, considerable variation has been reported in revision instructions and review durations for clinical research protocols and informed consent forms (ICFs) [8–10]. Such inter-committee variability undermines the consistency and transparency of the review process and often stems from differing interpretations of ethical standards and shortages in expert human resources. Delays in the review process can also postpone study initiation and, in turn, the introduction of new therapies and medical technologies. Moreover, if revision comments based on ethical standards

are inadequate or inappropriate, they may lead to insufficient or improper amendments to protocols and ICFs, potentially compromising participant safety.

Recent advances in large language models (LLMs), machine learning, deep learning, and generative artificial intelligence (AI) have shown promise in medical domains such as image analysis, electronic health record interpretation, and virtual health assistants [11–17]. LLMs, a type of deep learning-based generative AI, are capable of performing natural language processing (NLP) tasks [18]. ChatGPT, a widely known example of generative AI, has been trained on a large corpus of text and is capable of performing complex natural language tasks [19]. For instance, ChatGPT achieved scores at or above the passing threshold on publicly available questions from the United States Medical Licensing Examination (USMLE) [20] and the Japanese National Medical Licensing Examination [21,22], demonstrating its potential in high-level medical knowledge tasks.

Despite these promising capabilities, generative AI outputs are known to be prone to bias and hallucinations, raising ethical concerns [23–26]. In healthcare applications, AI transparency and reliability are crucial [24,25], and explainability is increasingly emphasized [23,26]. Explainable AI refers to technologies that enable AI to present the reasoning behind its outputs, thereby increasing the trustworthiness of the system.

Notably, few studies have examined the application of generative AI to the IRB review process itself. One key barrier is the high confidentiality of documents such as study protocols and ICFs, which makes it inappropriate to analyze them using cloud-based generative AI models (e.g., ChatGPT, Gemini, Copilot). Furthermore, IRB reviews typically involve complex deliberation by multiple experts with extensive experience and knowledge [1,27–29], and it remains unclear whether AI can adequately replicate such multifaceted processes.

Godwin et al. developed an AI tool to support the drafting of IRB application documents, seeking to improve the efficiency of document-preparation processes [30]. However, this approach is limited to assisting the creation of application materials and is not intended to support the process of reviewing submitted documents.

In contrast, the present project aims to develop an explainable AI tool to support the IRB document review process. Specifically, the system is designed to provide both the evaluation results—namely, which key elements the AI identified as critical for comprehensive IRB review, including research objectives and background, study design, participant-related risks and benefits, and other domain-specific considerations such as data management, statistical methodology, clinical expertise, and pharmacological perspectives—and the reasoning behind those evaluations. The goal is to assist IRB members in their decision-making and to improve the efficiency and consistency of the document review process. Importantly, this project does not aim to automate the IRB review; rather, it is based on the premise that AI should serve as a supporting tool for human reviewers.

The implementation of such AI support tools will eventually require advanced techniques such as retrieval-augmented generation (RAG) and fine-tuning (FT). However, it is first necessary to evaluate whether existing generative AI models can accurately and reproducibly extract key information from Japanese clinical research documents. Without this foundational capability, subsequent technological enhancements would lack reliability.

Accordingly, this article reports a foundational pilot study within our broader project. The overall structure of the project and the role of this study are illustrated in Fig 1.

This figure illustrates the broader framework for building an explainable AI tool to support document review in IRB processes. This pilot study serves as the first phase, evaluating the baseline accuracy and reproducibility of generative AI models in extracting and summarizing key components—namely, research objectives and background, research design, and participant-related risks and benefits—from clinical research protocols and ICFs.

In this study, we examined whether ChatGPT could accurately and reproducibly extract and summarize these key elements from Japanese-language documents. The findings are intended to serve as foundational data for the future implementation of RAG and FT technologies. Notably, this study does not evaluate the ability of AI to make judgments based on review criteria or to explain those judgments; such evaluation is planned in subsequent phases of the project.

**Target Documents:**
**Clinical Research Protocols / ICFs**

Written in Japanese

↓

**Pilot Phase (This Study)**
- GPT-4 / GPT-4o used for information extraction
- Comparison of custom vs standard prompts
- Key items: Study objectives, design, benefits/risks
- Evaluation of accuracy and reproducibility

↓

**Next Phase (Future Work)**
- Integration of Retrieval-Augmented Generation (RAG)
- Fine-tuning to improve output quality
- Enhancements: Multi-rater evaluation, quantuitative NLP metrics, API-based logging

↓

**Final Goal (Project Objective)**
- Develop an explainable AI tool to assist IRB members
- Support consistency, efficiency, and transparency in document review
- Supplement human decision-making (no automation intended)

**Fig 1. Project Overview and the Position of this Study in Developing an AI-based IRB Review Support System.**

We conducted two primary comparisons:

(1) a performance comparison between GPT-4 and GPT-4o;

(2) a reproducibility comparison between customized GPTs (modified via prompt engineering, without FT) and standard prompts.

## Results

In this study, we developed four customized GPT models (GPTs-1 to GPTs-4) using GPT-4 and GPT-4o to extract and summarize key components from Japanese clinical research protocols and ICFs. The target elements were: (1) research objectives and background, (2) research design, and (3) participant-related risks and benefits. We compared the accuracy and reproducibility of both models, and additionally evaluated the reproducibility of outputs generated using customized GPTs versus standard prompts.

## Accuracy evaluation

GPT-4o achieved 80% extraction accuracy for the "research objectives and background" element, and 100% extraction accuracy for both "research design" and "participant-related risks and benefits" (Table 1).

In contrast, GPT-4 achieved only 30% extraction accuracy for "research objectives and background" but demonstrated 100% extraction accuracy for the other two elements (Table 2).

## Reproducibility evaluation

Both models demonstrated high reproducibility, with at least 8 out of 10 outputs from each GPT rated as "consistent" or "partially consistent" (Table 3).

## Comparison between customized GPTs and standard prompts

Outputs generated using customized GPTs consistently produced exactly two well-structured participant-related benefit statements in each trial:

**Table 1. Accuracy of Information Extraction by GPT-4o: Extraction Accuracy with Research Protocol and ICF (10 Trials).**

| GPTs | Item | Accuracy | Partial Accuracy |
|------|------|----------|------------------|
| GPTs-1 | Research objectives and background | 80 | 20 |
| GPTs-2 | Research design | 100 | 0 |
| GPTs-3 | Participant-related risks and benefits (RP) | 100 | 0 |
| GPTs-4 | Participant-related risks and benefits (ICF) | 100 | 0 |

RP, research protocol; ICF, informed consent form.

This table presents the extraction accuracy and partial extraction accuracy rates across 10 trials for each GPT (GPTs-1 to GPTs-4). All values are shown as percentages.

**Table 2. Accuracy of Information Extraction by GPT-4: Extraction Accuracy with Research Protocol and ICF (10 Trials).**

| GPTs | Item | Accuracy (%) | Partial Accuracy (%) |
|------|------|--------------|----------------------|
| GPTs-1 | Research objectives and background | 30 | 70 |
| GPTs-2 | Research design | 100 | 0 |
| GPTs-3 | Participant-related risks and benefits (RP) | 100 | 0 |
| GPTs-4 | Participant-related risks and benefits (ICF) | 100 | 0 |

RP, research protocol; ICF, informed consent form.

This table presents the extraction accuracy and partial extraction accuracy rates across 10 trials for each GPT (GPTs-1 to GPTs-4). All values are shown as percentages.

**Table 3. Reproducibility of Information Extraction by GPT-4 and GPT-4o: Consistency Across 10 Trials.**

| GPTs | Item | GPT-4o | GPT-4 |
|------|------|--------|-------|
| GPTs-1 | Research objectives and background | High | High |
| GPTs-2 | Research design | High | High |
| GPTs-3 | Participant-related risks and benefits of research protocol | High | High |
| GPTs-4 | Participant-related risks and benefits of ICF | High | High |

This table summarizes the reproducibility results for each GPT (GPTs-1 to GPTs-4) based on 10 trials.

Reproducibility was defined as "high" when 8 or more trials were rated as "consistent" or "partially consistent."

Both GPT-4o and GPT-4 demonstrated equally high reproducibility across all tasks, with no notable differences observed between the models.

(1) Enhancement of information collection and follow-up care

(2) Expansion of treatment options

In contrast, outputs generated using standard prompts showed greater variability in both the number and type of extracted participant-related benefits, with categories and counts differing significantly across trials (Tables 4 and 5).

## Discussion

This pilot study evaluated the accuracy and reproducibility of ChatGPT models (GPT-4 and GPT-4o) in extracting and summarizing content from Japanese-language research protocols and ICFs. The aim was to assess the foundational capabilities of LLMs in preparation for their potential use in supporting IRB document review.

The results showed that GPT-4o achieved 80% extraction accuracy for the "research objectives and background" element and 100% extraction accuracy for both "research design" and "participant-related risks and benefits," demonstrating higher baseline accuracy than GPT-4. Notably, both models achieved perfect extraction accuracy for research design. However, some misinterpretations of medical terminology were observed. For example, "trastuzumab BS" was shortened to "trastuzumab," indicating inconsistencies in the handling of similar or abbreviated terms. These variations suggest that the model's interpretation of synonyms and abbreviations requires refinement, potentially through the application of RAG and FT.

Reproducibility was also high for both models, with 8 or more out of 10 outputs rated as "consistent" or "partially consistent" across all GPT variants. In particular, GPTs-3, which focused on extracting participant-related risks and benefits,

**Table 4. Number of Participant-Related Benefits Extracted by GPTs vs. Standard Prompts: Comparison Across 10 Trials Using GPT-4o.**

| Prompt Type | Number of trials | Extracted participant-related benefits | | | |
|---|---|---|---|---|---|
| | | Mean | SD | Max | Min |
| GPTs-3 | 10 | 2.0 | 0.0 | 2 | 2 |
| Standard prompts | 10 | 5.3 | 1.8 | 8 | 3 |

This table compares the mean, standard deviation, maximum, and minimum number of participant-related benefit statements extracted by GPTs-3 and standard prompts using GPT-4o.

**Table 5. Categories of Participant-Related Benefits Extracted by GPTs vs. Standard Prompts: Comparison Across 10 Trials Using GPT-4o.**

| Category | GPTs-3 | Standard prompts |
|---|---|---|
| Detailed information collection and follow-up care | Extracted | Extracted |
| Increase in treatment options | Extracted | Extracted |
| Reduction of economic burden | Not extracted | Extracted |
| Compensation for health risks | Not extracted | Extracted |
| Provision of personalized treatment | Not extracted | Extracted |
| Cost-effective treatment | Not extracted | Extracted |
| Evaluation of treatment efficacy and safety | Not extracted | Extracted |
| Collection and analysis of clinical data | Not extracted | Extracted |
| Protection of personal information | Not extracted | Extracted |
| Improvement in quality of life | Not extracted | Extracted |

This table presents the output categories of participant-related benefits statements generated by GPTs-3 and standard prompts. "Extracted" indicates that a given category appeared at least once during the 10 trials; "Not extracted" indicates that the category was never generated. GPTs-3, with clearly defined instructions and output formats, consistently extracted participant-related benefits explicitly described in the research protocol. In contrast, the outputs from standard prompts—due to their unstructured format—showed a wider and more diverse set of categories.

outperformed standard prompts by providing more accurate and consistent results. Specifically, the participant-related benefits of "improved follow-up care" and "expanded treatment options" were consistently extracted in all 10 trials.

By contrast, outputs generated using standard prompts showed greater variation in both the number and category of extracted participant-related benefits. This discrepancy is likely due to the effectiveness of prompt engineering, which explicitly defined the document sections to be referenced, the target elements to be extracted, and the output format. These findings highlight the critical role of prompt design in improving the accuracy and reproducibility of information extraction from Japanese-language documents. Based on these foundational performance results, further enhancement of AI models through RAG and FT is expected to support the practical implementation of AI tools for IRB review.

This study specifically assessed whether ChatGPT could extract and summarize critical information—namely, the research objectives and background, research design, and participant-related risks and benefits —that are essential for IRB document review. The findings demonstrated that current models possess sufficient baseline performance in terms of accuracy and reproducibility to support document-level review tasks. These results provide a basis for the future integration of advanced techniques such as RAG and FT.

Reproducibility and accuracy remain critical challenges for AI in healthcare. For example, research has highlighted the problem of inconsistent AI outputs across different datasets and conditions, presenting a barrier to clinical implementation [31]. The findings of this study align with such concerns, reinforcing that reproducibility and accuracy must be carefully balanced even in AI applications for document analysis.

From a technical perspective, we observed that ChatGPT struggled with PDF files that had complex structures or contained text stored as images. However, converting these documents to Word format and standardizing the heading structures significantly improved output stability. These results suggest that standardizing document formats is essential for ensuring accurate and reliable AI-assisted reviews.

**Limitations**

This study is positioned as a foundational pilot evaluation to assess the applicability of ChatGPT in supporting pre-reviews by IRBs, prior to the implementation of more advanced techniques such as RAG and FT.

The primary key performance indicators (KPIs) in this study were accuracy and reproducibility, focusing specifically on the quality of outputs extracting and summarizing the research objectives and background, research design, and participant-related risks and benefits. These KPIs were preliminarily established based on elements considered critical in IRB reviews; however, they should be further refined in future studies.

As a pilot study, the evaluation was conducted under practical constraints using manual review by a single assessor. This approach was chosen as a simple and pragmatic method for basic validation and was intended to provide foundational data for applying more advanced evaluation techniques in the next research phase.

Importantly, no formal benchmark—such as comparison with human IRB assessments—was applied at this stage. This limitation reflects the proof-of-concept nature of the study and will be addressed in the next phase of research.

Nevertheless, the following limitations must be acknowledged.

**1. Absence of gold standard reference data and unestablished evaluation criteria.** In this study, we manually compared ChatGPT's outputs with the original research protocol and ICF texts, assessing output accuracy and reproducibility based on predefined criteria.

Table 6 presents the detailed criteria for evaluating output accuracy, including content accuracy, semantic consistency, and expression consistency.

Table 7 outlines the classification standards for categorizing outputs into "consistent," "partially consistent," or "inconsistent" based on the predefined accuracy evaluation criteria.

Table 8 presents the evaluation criteria for reproducibility, categorizing outputs into high, moderate, or low reproducibility based on the number of trials rated as "consistent" or "partially consistent."

**Table 6. Criteria for Evaluating Output Accuracy.".**

| Item | Condition |
|---|---|
| Content accuracy | • Specific numerical values match perfectly.<br>• All major information related to research objectives and background is included.<br>• All major information related to research design and methods is included. |
| Semantic consistency | The meaning and intent of the original document are accurately conveyed. |
| Expression consistency | Even if phrasing or word choices differ, the meaning and intent remain unchanged (e.g., the number of patients increased" vs. "patient numbers increased"). |

This table defines the criteria used to assess the accuracy of GPT outputs compared with the research protocol and ICFs, encompassing content accuracy, semantic consistency, and expression consistency.

**Table 7. Classification Criteria for Output Accuracy.**

| Evaluation criteria | Description |
|---|---|
| Consistent | The output content matches the content of the research protocol completely and without error. |
| Partially consistent | The major information and intent of the output are mostly consistent with the research protocol. Minor discrepancies are observed, but these do not significantly impair the accuracy of the information, nor alter the meaning or intent. |
| Inconsistent | The output content does not match the major content of the research protocol, resulting in significant impairment of information accuracy or alteration of meaning or intent. |

This table outlines the standards for classifying GPT outputs into "consistent," "partially consistent," or "inconsistent" categories based on the predefined accuracy evaluation criteria.

**Table 8. Evaluation Criteria for Reproducibility.**

| Reproducibility level | Description |
|---|---|
| High | When 8 or more out of 10 trials are judged as "consistent" or "partially consistent" in line with Tables 7 and 8. |
| Moderate | When 5–7 out of 10 trials are judged as "consistent" or "partially consistent" in line with Tables 7 and 8. |
| Low | When 4 or fewer out of 10 trials are judged as "consistent" or "partially consistent" in line with Tables 7 and 8. |

This table classifies the reproducibility of GPT model outputs into three categories based on how many trials were rated as "consistent" or "partially consistent" (see Tables 7 and 8):

• ≥ 8 trials: High reproducibility.

• 5–7 trials: Moderate reproducibility.

• ≤ 4 trials: Low reproducibility.

## Limitations:

### • Lack of gold standard reference data:

Because publicly available IRB review results were not validated for correctness, we could not assess ChatGPT's accuracy through direct comparison with a verified reference set.

- **Preliminary KPIs and evaluation criteria:**

In this study, the evaluation criteria and KPIs were preliminarily established by focusing on key elements considered important in IRB document review—namely, the research objectives and background, research design, and participant-related risks and benefits.

The purpose was to assess the accuracy and reproducibility of extracting and summarizing these elements as a preliminary validation.

Since this study only targeted a limited set of elements, it remains necessary to further examine whether these KPIs are sufficient for broader IRB document review.

Because standardized benchmarks for such evaluations do not yet exist, future studies should aim to validate these KPIs and establish standardized evaluation methods.

- **Single evaluator and lack of inter-rater agreement:**

The evaluation was conducted by a single researcher, and inter-rater agreement was not assessed. Consequently, subjective judgments may have introduced bias in the results.

**Future directions:** To enhance objectivity and statistical validity, future research will adopt the following strategies:

- Inter-rater reliability analysis using multiple evaluators (e.g., Cohen's kappa);

- Quantitative NLP-based metrics (e.g., BLEU, ROUGE, Sentence-BERT similarity scores);

- Comparison with established review criteria (to assess alignment of AI outputs with ethical standards);

- Validation by IRB members.

These approaches will help develop appropriate evaluation frameworks for AI in IRB pre-review settings, especially when gold standard reference data are unavailable. In particular, directly comparing AI-generated outputs with assessments by experienced IRB reviewers will be essential for validating the practical utility of AI in ethical review, especially the degree to which AI tools can reflect the nuanced and context-sensitive judgments made by human experts.

2. **Unvalidated trial count.** In this proof-of-concept pilot study, 10 trials were conducted per task to assess output stability and reproducibility. This number was pragmatically selected to capture practical variation and trends in AI-generated outputs. The appropriateness of this trial count was not statistically validated, and no power analysis or convergence testing was conducted. This decision aligns with previous methodological recommendations for pilot studies, which emphasize feasibility testing and suggest that rigorous sample size calculations are not necessarily required in such contexts [32,33].

**Future directions:** Future studies will address this limitation through the following analyses:

- **Convergence analysis:** Evaluating output stability as the number of trials increases;

- **Confidence interval estimation:** Quantifying how variation in trial count affects reproducibility.

These analyses will inform the establishment of appropriate trial counts for evaluating AI-assisted IRB pre-reviews.

3. **Lack of API-based logging.** This study did not use the ChatGPT API, meaning that model usage (GPT-4 vs. GPT-4o), prompt details, execution timestamps, and output results were not automatically logged for each session. While the research team manually recorded the model used in each session and determined that this did not significantly affect the reliability of the results, the lack of automatic logs limits the objectivity, reproducibility, and transparency of the study.

In future studies, we plan to implement automated session logging using the API. This will enable systematic recording of model usage, prompt content, execution timing, and output results, thereby significantly improving the efficiency, completeness, and transparency of data collection and analysis.

**4. Exclusion of judgment-based review components from evaluation.** Finally, this study targeted the document-based IRB/CRB review process in Japan and was not intended to replace procedures requiring in-person assessment or site visits. When considering the application of AI to IRB processes in other countries, it will be necessary to examine whether AI can reasonably support judgment-based components of review—such as face-to-face meetings and site inspections—in addition to document-based assessments.

## Materials and methods

### Study overview

This study evaluated the accuracy and reproducibility of outputs generated by GPT models (OpenAI, San Francisco, CA, USA) in the document review of research protocols and ICFs. With a view toward future implementations of AI in supporting document-based IRB reviews in Japan, we focused on processing Japanese-language protocols using prompts written in Japanese.

Two types of comparisons were conducted:

1. a performance comparison between GPT-4 and GPT-4o (the latest model as of May 2024);

2. a reproducibility comparison between customized GPTs and standard prompts,

The primary KPIs were the accuracy and reproducibility of information extraction by AI for three elements:

1. research objectives and background;

2. research design;

3. participant-related risks and benefits.

These evaluation categories were preliminarily defined by focusing on three key elements considered important in IRB document review—namely, research objectives and background, research design, and participant-related risks and benefits.

This pilot study specifically assessed the accuracy and reproducibility of extracting and summarizing these limited elements.

Since the study targeted only a subset of information typically reviewed by IRB members, the KPIs used here are provisional and will be refined through future studies.

No FT or parameter adjustments were applied to the models. Instead, evaluation was conducted solely using prompt engineering with the default configuration provided by OpenAI.

### Data source

Mock review materials publicly available from the Ministry of Health, Labour and Welfare of Japan were used for this study [34]. These materials were created for training and education on IRB review and include fictional protocols and corresponding mock evaluation results across 10 disease domains [35–38].

Specifically, the dataset includes:

• Ten research projects covering oncology (*n* = 3), gastrointestinal diseases (*n* = 1), rheumatoid arthritis (*n* = 1), vaccines (*n* = 1), medical devices (*n* = 1), cardiovascular diseases (*n* = 1), pediatric psychiatric disorders (*n* = 1), and cognitive impairment (*n* = 1).

• Each set contains research protocols, explanatory documents, monitoring manuals, audit manuals, and expert reviewer reports (mock IRB assessments).

The validity of the mock evaluations is not disclosed, so the materials were used as reference standards, not as absolute ground truth.

For this study, we selected and analyzed the following protocol:

*"Phase II trial of trastuzumab for patients with advanced solid tumors without standard treatment options"* (Mock Review No. 1)

## Model selection and configuration

We evaluated the performance of the GPT-4 and GPT-4o models. GPT-4 is a conventional high-performance model with strong capabilities across a range of text generation tasks [39]. GPT-4o, released on May 13, 2024, offers twice the speed and 50% lower cost compared to GPT-4 Turbo. GPT-4o also achieves state-of-the-art performance in speech recognition, multilingual processing, and visual capabilities. In particular, improvements in multilingual capabilities allow for higher accuracy when processing non-English texts [40]. Given these characteristics, we considered it valuable to verify the accuracy improvements offered by GPT-4o.

We did not apply FT or model parameter adjustments. Instead, we used the GPTs feature provided by OpenAI [41] to develop customized GPTs tailored to extract specific categories of information (Table 9).

## Evaluation process and comparison

**Procedure for GPTs trials.** To assess accuracy and reproducibility, the following procedure was implemented:

1. **Start a new chat session**

   Each trial was initiated in a new session to eliminate any influence from previous interactions.

2. **Select model and GPT**

   GPT-4 or GPT-4o was selected along with one of the four customized GPTs (GPTs-1 to GPTs-4).

3. **Upload document**

   The target research protocol or ICF was uploaded, then outputs were automatically generated using the embedded prompts.

**Table 9. Overview of the Four Customized GPTs and Their Prompt Structures.**

| GPTs | Overview of prompts |
| --- | --- |
| GPTs-1 | Accurately extract the research objectives and background from the provided research protocol. |
| GPTs-2 | Accurately extract details such as target patients, trial phase, trial type, randomization, control group setup, blinding, and sample size. |
| GPTs-3 | Accurately extract the participant-related risks and benefits for the subjects from sections 16.3.1 and 16.3.2 of the research protocol. |
| GPTs-4 | Accurately extract the participant-related risks and benefits for the subjects from section 6 of the ICF. |

This table summarizes the four customized GPTs (GPTs-1 to GPTs-4) developed for information extraction from research protocols and informed consent forms (ICFs). Each GPT was designed to target a specific document type and extraction goal aligned with one of the evaluation categories: research objectives and background, research design, and participant-related risks and benefits.

4. **Save output**

Each result was saved with a name format such as "GPTs_name_trialNumber." All outputs were evaluated after completing 10 trials for each GPT.

5. **Repeat process**

Each GPT was run 10 times with both models, totaling 80 trials.
**Procedure for standard prompt trials.** For comparison, we also conducted evaluations using standard prompts.

1. **Start a new chat session**

Each trial was conducted in an independent session.

2. **Model selection**

The GPT-4o model was used.

3. **Upload document and enter prompt**

The research protocol or ICF was uploaded. The following prompt (in Japanese) was entered:
"You are an IRB reviewer. Please extract the participant-related benefits for participants from the following research protocol."

4. **Save output**

Each result was saved as "StandardPrompt_trialNumber" and evaluated after all 10 trials were completed.

5. **Repeat process**

A total of 10 trials were performed.

## Evaluation criteria for accuracy and reproducibility

All outputs were manually reviewed by one evaluator based on predefined criteria (Tables 6–8) using the following process:

1. **Accuracy.** Outputs were classified as "consistent," "partially consistent," or "inconsistent" using the definitions presented in Tables 6 and 7.

2. **Reproducibility.** Reproducibility was determined based on the criteria presented in Table 8, using the number of trials rated as "consistent" or "partially consistent" per model (out of 10).

The extracted text and its translation, GPT-generated outputs, results from standard prompts, and the mock IRB evaluation form are provided as supporting information (S1 to S11 Files) to ensure transparency and reproducibility.

## Supporting information

**S1 File. List of extracted sections from the research protocol and ICF (Japanese original and English translation).** This file contains the original Japanese text and English translations of the specific sections from the research protocol and informed consent form (ICF) that were subject to extraction and summarization in this study.
(XLSX)

**S2 File. GPTs-1 output results (GPT-4o model).** A list of outputs generated by GPTs-1 using the GPT-4o model, focusing on the extraction of the "research objectives and background" element.
(XLSX)

**S3 File. GPTs-1 output results (GPT-4 model).** A list of outputs generated by GPTs-1 using the GPT-4 model.
(XLSX)

**S4 File. GPTs-2 output results (GPT-4o model).** A list of outputs generated by GPTs-2 using the GPT-4o model, focusing on extraction of the "research design" element.
(XLSX)

**S5 File. GPTs-2 output results (GPT-4 model).** A list of outputs generated by GPTs-2 using the GPT-4 model.
(XLSX)

**S6 File. GPTs-3 output results (GPT-4o model).** A list of outputs generated by GPTs-3 using the GPT-4o model, focusing on extraction of the "participant-related risks and benefits" from the research protocol.
(XLSX)

**S7 File. GPTs-3 output results (GPT-4 model).** A list of outputs generated by GPTs-3 using the GPT-4 model.
(XLSX)

**S8 File. GPTs-4 output results (GPT-4o model).** A list of outputs generated by GPTs-4 using the GPT-4o model, focusing on extraction of the "participant-related risks and benefits" from the ICF.
(XLSX)

**S9 File. GPTs-4 output results (GPT-4 model).** A list of outputs generated by GPTs-4 using the GPT-4 model.
(XLSX)

**S10 File. Output results using a standard prompt (GPT-4o model).** This file contains the outputs generated using a standard prompt in GPT-4o, used for comparison with GPTs-based outputs.
(XLSX)

**S11 File. Mock IRB evaluation form (Japanese only).** This is a publicly available mock IRB evaluation form published by the Ministry of Health, Labour and Welfare of Japan. It was referenced in determining the provisional KPIs used in this study. Please note that the correctness of the evaluation results in this form has not been publicly verified, and thus cannot be directly compared with the AI-generated outputs.
(PDF)

## Acknowledgments

We would like to express our gratitude to Associate Professor Kenji Hirata from the Department of Diagnostic Imaging, Graduate School of Medicine, Hokkaido University, for providing valuable insights related to this project. We also thank Professor Shiro Hinotsu from the Division of Biostatistics and Data Management, Sapporo Medical University, for dedicating his time to discussing this topic with us.

We thank ChatGPT (OpenAI) for assisting with the English translation of Japanese-language research materials, including manuscript drafts, mock protocols, and output examples. All AI-generated translations were carefully reviewed and verified by the authors. We also thank Editage (Cactus Communications) for providing final English proofreading.

The authors take full responsibility for the accuracy and integrity of the manuscript content.

## Author contributions

**Conceptualization:** Naoki Nishimoto, Yoichi M. Ito.

**Data curation:** Yasuko Fukataki, Wakako Hayashi.

**Formal analysis:** Yasuko Fukataki.

**Investigation:** Yasuko Fukataki.

**Methodology:** Yasuko Fukataki.

**Supervision:** Yasuko Fukataki, Yoichi M. Ito.

**Validation:** Yasuko Fukataki.

**Visualization:** Yasuko Fukataki.

**Writing – original draft:** Yasuko Fukataki.

**Writing – review & editing:** Wakako Hayashi, Naoki Nishimoto, Yoichi M. Ito.

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
