## [Decision Letter · Decision Letter 0]

PDIG-D-24-00501Ethical review of clinical research with generative AI: Evaluating ChatGPT's accuracy and reproducibilityPLOS Digital Health Dear Dr. Fukataki, Thank you for submitting your manuscript to PLOS Digital Health. After careful consideration, we feel that it has merit but does not fully meet PLOS Digital Health's publication criteria as it currently stands. Therefore, we invite you to submit a revised version of the manuscript that addresses the points raised during the review process. Please submit your revised manuscript within 60 days Mar 30 2025 11:59PM. If you will need more time than this to complete your revisions, please reply to this message or contact the journal office at digitalhealth@plos.org. Please include the following items when submitting your revised manuscript:* A rebuttal letter that responds to each point raised by the editor and reviewer(s). You should upload this letter as a separate file labeled 'Response to Reviewers '. This file does not need to include responses to any formatting updates and technical items listed in the 'Journal Requirements' section below.* A marked-up copy of your manuscript that highlights changes made to the original version. You should upload this as a separate file labeled 'Revised Manuscript with Track Changes '.* An unmarked version of your revised paper without tracked changes. You should upload this as a separate file labeled 'Manuscript '. If you would like to make changes to your financial disclosure, competing interests statement, or data availability statement, please make these updates within the submission form at the time of resubmission. Guidelines for resubmitting your figure files are available below the reviewer comments at the end of this letter. We look forward to receiving your revised manuscript. Kind regards, Po-Chih Kuo, Ph. D.Section EditorPLOS Digital Health Po-Chih KuoSection EditorPLOS Digital Health Leo Anthony CeliEditor-in-ChiefPLOS Digital Healthorcid.org/0000-0001-6712-6626 **Additional Editor Comments (if provided):****Reviewers' Comments:** Reviewer's Responses to Questions

**Comments to the Author**

1. Does this manuscript meet PLOS Digital Health’s publication criteria ? Is the manuscript technically sound, and do the data support the conclusions? The manuscript must describe methodologically and ethically rigorous research with conclusions that are appropriately drawn based on the data presented.

Reviewer #1: Partly

Reviewer #2: Partly

2. Has the statistical analysis been performed appropriately and rigorously?

Reviewer #1: No

Reviewer #2: N/A

3. Have the authors made all data underlying the findings in their manuscript fully available (please refer to the Data Availability Statement at the start of the manuscript PDF file)?

Reviewer #1: Yes

Reviewer #2: No

4. Is the manuscript presented in an intelligible fashion and written in standard English?

Reviewer #1: Yes

Reviewer #2: Yes

5. Review Comments to the Author

Reviewer #1: This manuscript explores the application of GPT-4 and GPT-4o in ethical reviews of clinical research protocols, with a focus on Japanese-language documents. It demonstrates promising advancements in AI-driven review processes, however, there are notable gaps in methodological clarity, scope, and alignment with the stated objectives;

1.Consider integrating the following reference into your work: Godwin, R. C., et al. (2024). IRB-draft-generator: A generative AI tool to streamline the creation of institutional review board applications. SoftwareX, 25, 101601.

2.The study highlights the accuracy of GPT models but does not address the potential for task-specific fine-tuning. One of the provided inaccuracies emanated from medical terminology; this could be mitigated/reduced by fine-tuning domain-specific datasets. Including this as a future direction would strengthen the study's practical implications. Moreso, more samples that reflect the inaccuracies from the evaluated GPT’s could be provided /attached.

3.One critical limitation is the inability of generative AI to replicate in-person site visits, which are sometimes essential for IRB reviewers to understand study contexts. This limitation should be explicitly discussed in the manuscript.

4.The methodology lacks details on how custom GPTs were developed. Were additional datasets used, and if so, what were their size and origin? If not, then, a clear comparison between custom GPTs and standard GPTs on identical tasks would provide a stronger basis for evaluating the added value of customization.

5.The manuscript does not clearly explain the basis of 10 iterations as it only assigns an evaluation text (consistent value) based on runs. This can be debatable since there is no standard evaluation criteria. This could also varry depending on the context of the sentences, and structure. Paraphrasing the sentences could also give us different results.

Hence, the evaluation does not reflect some statistical or practical rationale. Discussing whether the results remain consistent with fewer or more runs, or providing a maximum or minimum runs per any sentence in the same medical domain/subdomain would improve reproducibility and transparency.

6.While the study focuses on a single research protocol, the data source reportedly included multiple protocols. Testing GPT models on another protocol would enhance the generalizability of findings and should be considered in future work.

7.What benchmark or baseline was used to evaluate the models? Including a comparison with human reviewers or protocols in a similar language (e.g., English) would provide a clearer context for the reported accuracy and reproducibility rates. In case there is absence of a benchmark, an ablation study could be opted for or multiple but divergent experiments could be carried out.

8.While the manuscript addresses AI accuracy and reproducibility, it provides limited discussion on ethics in clinical research, which is highlighted in the title. The aurthos could either expand the discussion to include ethical considerations or revise the title to better reflect the study's actual focus.

9.Including a sample review form would help clarify the scope and nature of the analysis and also provide readers with a better understanding of the alignment between GPT outputs and IRB expectations.

10.The authors should avoid repetitive descriptions, particularly regarding GPT-4o’s performance consistency. Consolidating these points will improve readability.

Reviewer #2: The manuscript investigates the application of ChatGPT, a generative AI model, in conducting ethical reviews of clinical research. It evaluates the tool’s accuracy, reproducibility, and potential role in automating aspects of ethical review processes. The study is timely and addresses an innovative area in digital health, demonstrating the potential for AI to support or augment traditional human-driven processes. However, several aspects of the manuscript require clarification, additional detail, or revisions to meet the publication criteria.

Detailed Feedback

1. Data Availability:

• The manuscript does not fully satisfy PLOS Digital Health’s data availability requirements. While summary statistics are provided, the underlying raw data and detailed analysis files are not available.

• Recommendation: Deposit the dataset in a public repository and include access details (e.g., a DOI). If restrictions on sharing exist (e.g., participant privacy or third-party data restrictions), these must be explicitly stated in the Data Availability Statement.

2. Methodological Rigor:

• The manuscript describes a methodology for evaluating ChatGPT’s accuracy and reproducibility, but some critical details are missing:

• Statistical metrics such as inter-rater reliability or confidence intervals for key findings are absent.

• The process for defining accuracy and reproducibility thresholds is not well-articulated.

• Recommendation: Provide more detailed descriptions of the evaluation framework, particularly regarding statistical validation, to enhance the methodological rigor.

3. Reproducibility:

• The study’s reproducibility is hindered by limited information about the dataset and evaluation criteria.

• Recommendation: Include sufficient detail on the dataset used, the evaluation framework, and the specific metrics or benchmarks applied to assess ChatGPT’s performance. This will enable replication of the study by other researchers.

4. Ethical Considerations:

• While the manuscript focuses on the use of AI in ethical reviews, it does not address potential biases, hallucinations, or ethical concerns arising from the use of generative AI in this context.

• Recommendation: Expand the discussion to include potential risks of bias, misuse, and hallucinations in AI-generated outputs. Consider strategies to mitigate these risks in practice.

5. KPIs and Performance Metrics:

• The manuscript provides key performance indicators (KPIs), but they appear preliminary and lack justification.

• Recommendation: Clearly state that the KPIs are tentative and discuss how they could be refined in future research.

6. Figures and Tables:

• While useful, the figures and tables lack detailed captions and self-contained explanations.

• Recommendation: Add detailed captions to all figures and tables to ensure they can stand alone without requiring extensive cross-referencing to the main text.

7. Writing and Presentation:

• The manuscript is written in standard English and is generally well-organized. However, some sections (e.g., the Abstract and Discussion) include repetitive language and overly technical terms that may limit accessibility.

• Recommendation: Streamline the Abstract and Discussion to eliminate redundancy. Simplify overly technical language to ensure broader accessibility without losing depth.

Minor Concerns

1. Research Ethics: Confirm that the study complies with ethical standards for AI research, particularly regarding transparency and accountability in dataset usage.

2. Potential Conflicts of Interest: Address any potential conflicts of interest that may arise from the authors’ affiliations or funding sources.

Strengths of the Manuscript

• Innovation: The manuscript explores a novel application of generative AI in an important domain.

• Timeliness: The study addresses an area of growing interest in digital health and AI ethics.

• Clarity: Despite some gaps, the manuscript is generally well-written and organized.

Conclusion

The manuscript has significant potential to contribute to digital health research. However, revisions are necessary to address issues with data availability, methodological rigor, and reproducibility. By incorporating these changes, the study can provide a robust and impactful exploration of ChatGPT’s role in clinical ethical reviews.

6. PLOS authors have the option to publish the peer review history of their article (what does this mean? ). If published, this will include your full peer review and any attached files.

**Do you want your identity to be public for this peer review?** For information about this choice, including consent withdrawal, please see our Privacy Policy .

Reviewer #1: No

Reviewer #2: No

---

## [Decision Letter · Decision Letter 1]

Developing artificial intelligence tools for institutional review board pre-review: A pilot study on ChatGPT’s accuracy and reproducibility

PDIG-D-24-00501R1

Dear Ms. Fukataki,

We are pleased to inform you that your manuscript 'Developing artificial intelligence tools for institutional review board pre-review: A pilot study on ChatGPT’s accuracy and reproducibility' has been provisionally accepted for publication in PLOS Digital Health.

Best regards,

Po-Chih Kuo, Ph. D.

Section Editor

PLOS Digital Health

**Additional Editor Comments (if provided):**

**Reviewer Comments (if any, and for reference):**

Reviewer's Responses to Questions

**Comments to the Author**

1. If the authors have adequately addressed your comments raised in a previous round of review and you feel that this manuscript is now acceptable for publication, you may indicate that here to bypass the “Comments to the Author” section, enter your conflict of interest statement in the “Confidential to Editor” section, and submit your "Accept" recommendation.

Reviewer #1: All comments have been addressed

2. Does this manuscript meet PLOS Digital Health’s publication criteria ? Is the manuscript technically sound, and do the data support the conclusions? The manuscript must describe methodologically and ethically rigorous research with conclusions that are appropriately drawn based on the data presented.

Reviewer #1: Yes

3. Has the statistical analysis been performed appropriately and rigorously?

Reviewer #1: Yes

4. Have the authors made all data underlying the findings in their manuscript fully available (please refer to the Data Availability Statement at the start of the manuscript PDF file)?

Reviewer #1: Yes

5. Is the manuscript presented in an intelligible fashion and written in standard English?

PLOS Digital Health does not copyedit accepted manuscripts, so the language in submitted articles must be clear, correct, and unambiguous. Any typographical or grammatical errors should be corrected at revision, so please note any specific errors here.

Reviewer #1: Yes

6. Review Comments to the Author

Please use the space provided to explain your answers to the questions above. You may also include additional comments for the author, including concerns about dual publication, research ethics, or publication ethics. (Please upload your review as an attachment if it exceeds 20,000 characters)

Reviewer #1: The authors have appropriately framed the study, acknowledged its limitations comprehensively, and outlined a clear path for future research, thus addressing my original comments and suggestions. This is also on the basis that this is among a few pilot studies in this core area of research. We are looking forward to future works that extend this pilot.

7. PLOS authors have the option to publish the peer review history of their article (what does this mean? ). If published, this will include your full peer review and any attached files.

**Do you want your identity to be public for this peer review?** For information about this choice, including consent withdrawal, please see our Privacy Policy .

Reviewer #1: **Yes: ** Richard Kimera
